# GaussianFusion: Unified 3D Gaussian Representation for Multi-Modal Fusion Perception

**Xiao Zhao,**[*] **Chang Liu,**[*] **Mingxu Zhu, Zheyuan Zhang, Linna Song, Qingliang Luo, Chufan Guo** & **Kuifeng Su**
Tencent, Autonomous Driving Lab
Beijing, 100000, CN
{shawzhao,changeliu}@tencent.com

## Abstract

The bird's-eye view (BEV) representation enables multi-sensor features to be fused within a unified space, serving as the primary approach for achieving comprehensive 3D perception. However, the discrete grid representation of BEV leads to significant detail loss and limits feature alignment and cross-modal information interaction in multimodal fusion perception. In this work, we break from the conventional BEV paradigm and propose a new universal framework for multimodal fusion based on 3D Gaussian representation. This approach naturally unifies multi-modal features within a shared and continuous 3D Gaussian space, effectively preserving edge and fine texture details. To achieve this, we design a novel forward-projection-based multi-modal Gaussian initialization module and a shared cross-modal Gaussian encoder that iteratively updates Gaussian properties based on an attention mechanism. GaussianFusion is inherently a task-agnostic model, with its unified Gaussian representation naturally supporting various 3D perception tasks. Extensive experiments demonstrate the generality and robustness of GaussianFusion. On the nuScenes dataset, it outperforms the 3D object detection baseline BEVFusion by 2.6 NDS. Its variant surpasses GaussFormer on 3D semantic occupancy with 1.55 mIoU improvement while using only 30% of the Gaussians and achieving a 450% speedup.

## 1 Introduction

Fusing complementary signals captured by different sensors is essential for autonomous driving perception systems. Leveraging the distinct characteristics of each sensor helps reduce prediction uncertainty, leading to more accurate and robust perception outcomes (Liu et al., 2023b; Bai et al., 2022; Yan et al., 2023). Since different sensors present data in varying formats, such as cameras providing perspective semantic data and Lidar capturing 3D spatial information, multi-modal fusion faces significant challenges due to these view discrepancies. To address this, some methods (Vora et al., 2020; Bai et al., 2022; Li et al., 2024; Wang et al., 2024b) achieve multi-modal 3D object detection through point-level fusion. However, point-level fusion strategies are generally unsuitable for semantics-oriented 3D perception tasks like 3D semantic occupancy prediction. Consequently, recent approaches aim to construct unified representations for multi-modal feature fusion.

Recently, the shared Bird's Eye View (BEV) space has emerged as a promising direction for fusing cross-modal features to enable task-agnostic learning. Several existing fusion methods (Liu et al., 2023b; Wang et al., 2023a), such as BEVFusion (Liu et al., 2023b), integrate multimodal information via CNNs and feature concatenation, while MetaBEV (Ge et al., 2023) mitigates

Table 1: Impact of BEV size on model performance

| Method | BEV size | Grid size | Memory↓ | NDS↑ |
|---|---|---|---|---|
| BEVFusion | 100×100 | 1.008m | **3228** M | 70.5 |
| | 200×200 | 0.504m | 5140 M | 71.4 |
| | 400×400 | 0.252m | 20560 M | 72.7 |
| GaussianFusion | 100×100 | 1.008m | 3576 M | 73.1 |
| | 200×200 | 0.504m | 5418 M | 74.0 |
| | 400×400 | 0.252m | 6151 M | **74.4** |

---

[*]Equal contribution.

cross-modal feature misalignment by introducing meta-BEV queries. However, despite its widespread adoption in multimodal 3D perception, the BEV representation inherently suffers from limitations in information expression.

BEV directly discretizes and quantizes data, leading to inevitable information loss. During feature extraction, perception data are projected onto a fixed-resolution BEV grid, which compresses spatial information. This issue becomes particularly severe when the BEV resolution is low, as it directly impacts model performance by failing to adequately preserve fine-grained scene structures. While increasing the BEV resolution will bring unacceptable computational overhead, as shown in Table 1. Additionally, BEV fusion strategies often rely on simple feature concatenation or weighted summation, which are insufficient for effective cross-modal feature interaction and alignment, ultimately leading to suboptimal fusion performance, as illustrated in Fig. 1(a).

To address these challenges, we introduce a fusion approach based on 3D Gaussian Splatting (3DGS) (Kerbl et al., 2023) to achieve more fine-grained information modeling and more natural multimodal alignment. As shown in Fig. 1(b), 3DGS employs continuous Gaussian distributions to represent the scene, preserving rich geometric and semantic information in the Gaussian stage and preventing the early quantization-induced information loss seen in

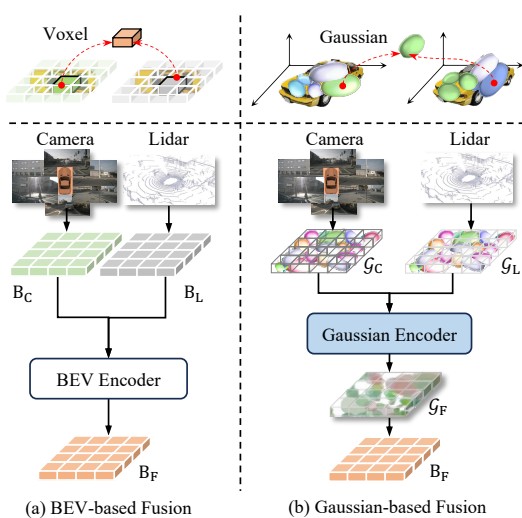

Figure 1: Comparison of the discrete BEV representation fusion paradigm (Liu et al., 2023b) and our proposed continuous Gaussian representation fusion paradigm. B, $\mathcal{G}$, C, L, and F denote BEV, Gaussian, Camera, Lidar, and Fusion.

BEV-based methods. Unlike direct BEV quantization, 3DGS aggregates information before its final projection onto the BEV grid, allowing cross-modal features to interact at a higher-dimensional level and capturing finer spatial structures prior to quantization, Table 1 shows the effectiveness of this strategy. Moreover, the covariance matrices of Gaussians enable adaptive modeling of uncertainty, enhancing the representation of object shapes and boundaries.

Specifically, inspired by (Philion & Fidler, 2020), we propose a forward projection Gaussian initialization strategy to better initialize camera Gaussian representations in 3D space rather than using random initialization (H et al., 2024). To further achieve continuous alignment and cross-modal feature enhancement, we construct a shared Gaussian encoder. The shared Gaussian encoder supports cross-feature learning of 3D Gaussian features from different modalities, where the covariance matrix of each 3D Gaussian adaptively captures feature differences between modalities and iteratively updates the Gaussian parameters. Camera and LiDAR Gaussians are naturally fused via a Gaussian mixture model, and a high-performance Gaussian-to-voxel fusion module aggregates surrounding Gaussians to generate voxel features, enabling task-agnostic 3D perception. We conduct extensive experiments on BEV object detection and 3D occupancy prediction tasks to validate the generality of the GaussianFusion. Main contributions are as follows:

- We propose the first unified 3D Gaussian representation multi-modal fusion framework, where cross-view and cross-modal Gaussian representations are naturally aggregated through the Gaussian mixture model.

- A progressive update strategy is designed to optimize the multi-modal 3D Gaussian properties iteratively.

- The shared 3D Gaussian encoder enables alignment and complementary enhancement of cross-modal features, allowing Gaussian representations from both modalities to achieve consistent uncertainty within a unified space.

- Our GaussianFusion achieves state-of-the-art benchmarks in task-agnostic methods on various 3D perception tasks within the nuScenes dataset.

## 2 RELATED WORK

### 2.1 MULTIMODAL 3D PERCEPTION

We categorize current multimodal fusion methods into object-centric methods and dense BEV methods. Object-centric methods (Vora et al., 2020; Chen et al., 2023; Zhou & T, 2023; Yin et al., 2024; Li et al., 2024; Wang et al., 2024b) are specifically designed for tasks such as 3D object detection or tracking. Advanced object-centric (Li et al., 2024; Wang et al., 2024b) methods typically use 2D detection results on camera to enhance multi-modal fusion 3D detection. Additionally, some works (Yang et al., 2022; Yan et al., 2023) use query-based 3D detection decoders to learn features from perspective images and Lidar BEV features directly. However, these object-centric methods cannot easily generalize to dense semantic tasks such as BEV map segmentation and 3D occupancy prediction. Dense BEV methods (Li et al., 2022a; Liu et al., 2023b; Liang et al., 2022; Chen et al., 2022; Zhao et al., 2024b; Ge et al., 2023; Jiao et al., 2023; Wang et al., 2023a) naturally adapt to various tasks. Both BEVFusion (Liu et al., 2023b) and UniTR(Wang et al., 2023a) achieve multi-modal BEV fusion perception through CNN and feature concatenation in the BEV space. In addition, MetaBEV (Ge et al., 2023) proposed a learnable cross-attention mechanism to generate unified BEV features. BEV or 3D voxel also provides a unified representation for 3D occupancy prediction. Some methods design fusion modules (Pan et al., 2024; Wang et al., 2023c; Ming et al., 2024) based on voxel, such as adaptive fusion (Wang et al., 2023c), etc. for multi-modal 3D occupancy prediction. There are also many camera-only methods (Zhao et al., 2024a; Wang et al., 2024a; Li et al., 2023; Lu et al., 2023; Tian et al., 2024; Li et al., 2025; Ma et al., 2024b; Cao et al., 2024; Liu et al., 2024; Ma et al., 2024a) for 3D semantic occupancy prediction based on voxel representation. However, discrete voxel representations may result in significant detail loss and hinder effective multimodal complementary fusion.

### 2.2 3D GAUSSIAN SPLATTING

3D Gaussian Splatting (3DGS) (Kerbl et al., 2023) combines the advantages of implicit neural radiance fields (Mildenhall et al., 2021) and voxel-based explicit radiance fields (Fridovich-Keil et al., 2022; Müller et al., 2022) and is widely applied in 3D reconstruction of 2D image. 3DGS uses a set of Gaussian functions to capture the geometric shapes and semantic of different objects or regions within a scene, effectively representing the scene. Based on this, some works (Ye et al., 2025; Hu et al., 2024) leverage 3DGS's multi-view synthesis capabilities to achieve 3D scene segmentation.

Recent studies (Gan et al., 2024; Chabot et al., 2024; H et al., 2024; Zuo et al., 2024; Liu et al., 2025) have applied 3DGS to vision-only 3D semantic occupancy prediction, BEV segmentation, and end-to-end autonomous driving. However, these methods rely on randomly initialized Gaussians. For example, GaussianFormer (H et al., 2024) randomly initializes and re-predicts Gaussian parameters in each iteration, making fine-tuning difficult and limiting accurate scene representation. Moreover, these methods do not fully exploit the advantages of Gaussian Mixture Models (GMM) for seamless multimodal Gaussian fusion. In contrast, we propose a forward-projection-based 3DGS parameter initialization and a shared optimization model, leveraging GMM to fuse multi-view camera and LiDAR features within a shared space, ultimately enabling dense semantic understanding and object-centric multitask perception.

## 3 METHODS

### 3.1 OVERALL ARCHITECTURE

The overall pipeline of GaussianFusion is illustrated in Fig. 2, with the goal of fusing multimodal features through 3D Gaussian representations, which naturally preserve both geometric and semantic information. We first initialize separate 3D Gaussian representations for camera and Lidar, denoted as $\mathcal{G}_c \sim Q_c$ and $\mathcal{G}_L \sim Q_L$, within a unified space. Then, the multimodal Gaussian sets are processed through a shared Gaussian Encoder, enabling the integration of semantic and geometric information

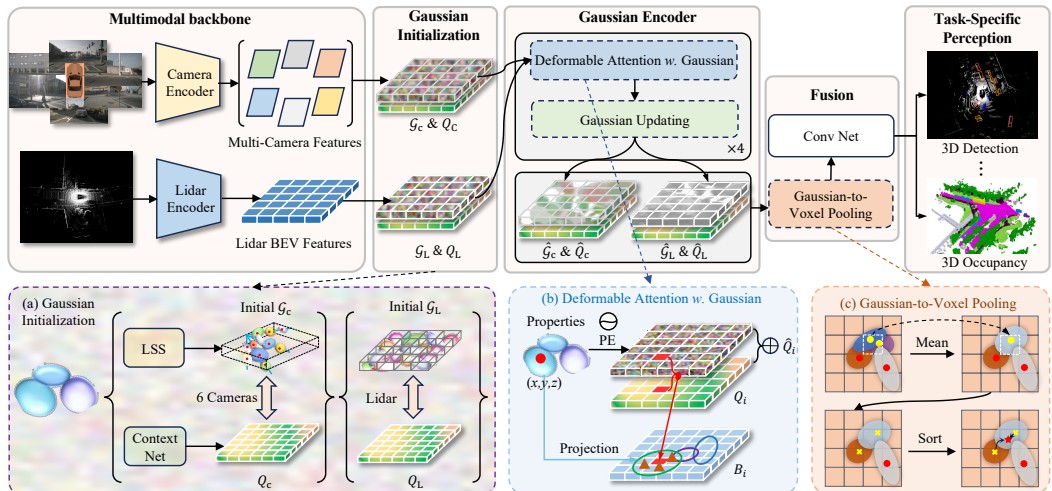

Figure 2: Overview of the GaussianFusion framework. Initial Gaussians are refined by a shared encoder and fused in Gaussian space, followed by task-specific heads for 3D perception.

from both modalities. Finally, the learned 3D Gaussian sets $\hat{\mathcal{G}}_c \sim \hat{Q}_c$ and $\hat{\mathcal{G}}_L \sim \hat{Q}_L$ are fused within the unified Gaussian space and fed into task-specific heads to perform 3D perception.

## 3.2 GAUSSIAN INITIALIZATION

BEVFusion (Liu et al., 2023b) projects multimodal features into discrete BEV space for fusion. In contrast, inspired by the Gaussian mixture model paradigm of 3DGS (Kerbl et al., 2023) for modeling scene geometry and semantics, we utilize two Gaussian sets, $\mathcal{G}c$ and $\mathcal{G}_L$, within a shared space to represent surround-view camera and Lidar information, respectively, to achieve a seamless multimodal fusion.

**Camera Gaussian Initialization with Forward Projection.** The properties of every single 3D Gaussian function are defined by a mean $\boldsymbol{\mu} \in \mathbb{R}^3$, scale $\mathbf{s} \in \mathbb{R}^3$, and rotation vectors $\mathbf{r} \in \mathbb{R}^4$. Given $N$ surround camera view, each camera's view can be represented by a set of 3D Gaussian distributions $G_{c,i} \in \mathbb{R}^{D_g \times D_c \times H_c \times W_c}|i = 1, 2, \cdots, N$, where $D_g = \boldsymbol{\mu} + \mathbf{s} + \mathbf{r}$, $D_c$ denotes the number of discrete depths for camera, as shown in Fig. 2(a), a Gaussian function is assigned to a depth point. It is worth noting that this is completely different from GausianFormer (H et al., 2024), which randomly initializes a set of Gaussians in space, which makes model optimization more difficult. Specifically, inspired by (Huang & Huang, 2022b;a; Liu et al., 2023b; Philion & Fidler, 2020), given surround camera input features $F_{c,i} \in \mathbb{R}^{C \times H_c \times W_c}, i = 1, 2, \cdots, N$, where $C$, $H_c$ and $W_c$ represent the channel, height, and width of the camera features. $F_{c,i}$ is fed to LSS (Philion & Fidler, 2020) to obtain the depth distribution $D_i \in \mathbb{R}^{D_c \times H_c \times W_c}, i = 1, 2, \cdots, N$, $D_i$ are then used as the initial mean $\boldsymbol{\mu}$ of the Gaussian, which is the location of every single Gaussian center. For $\mathbf{s}$ and $\mathbf{r}$, we initialize them randomly. And $\mathcal{G}_c = \{G_{c,i}\}, \mathcal{G}_c \in \mathbb{R}^{N \times D_g \times D_c \times H_c \times W_c}$.

The image features $F_{c,i}$ are processed through a context network composed of multiple convolutional layers to obtain the semantic features $F'_{c,i}$. Next, an inner product is computed between $D_i$ and $F'_{c,i}$ to derive the features at each depth point in 3D space, denoted as initial query features $Q_c = \{Q_{c,i} \in \mathbb{R}^{C \times D_c \times H_c \times W_c}|i = 1, 2, \cdots, N\}$. Then, $Q_c$ are associated with Gaussian ($\mathcal{G}_c \sim Q_c$). For a given Gaussian set ($g_c \sim q_c, g_c \in \mathcal{G}_c, q_c \in Q_c$), the feature at a point $\mathbf{p} = (x, y, z)$ within its elliptical space are:

$$g_c(\mathbf{p}; \boldsymbol{\mu}, \mathbf{s}, \mathbf{r}) = \exp\left(-\frac{1}{2}(\mathbf{p} - \boldsymbol{\mu})^T \boldsymbol{\Sigma}^{-1}(\mathbf{p} - \boldsymbol{\mu})\right) q_c, \tag{1}$$

where $\boldsymbol{\Sigma} = \mathbf{RSS}^T\mathbf{R}^T$, $\mathbf{S} = \mathrm{diag}(\mathbf{s})$, and $\mathbf{R} = \mathrm{q2r}(\mathbf{r})$. $\boldsymbol{\Sigma}, \mathrm{diag}(\cdot)$, and $\mathrm{q2r}(\cdot)$ represent the covariance matrix, the function that constructs a diagonal matrix from a vector, and the function that transforms a quaternion into a rotation matrix, respectively.

**Lidar Gaussian initialization.** Lidar's BEV space naturally provides an initialization for the Gaussian mean $\boldsymbol{\mu}$. And, $\mathbf{s}$ and $\mathbf{r}$ are initialized randomly. The Gaussian initialization for Lidar is formulated as $\mathcal{G}_L \in \mathbb{R}^{C \times H_L \times W_L}$. Since directly extracting information from the massive raw lidar point clouds to construct 3D Gaussian representations is both difficult and computationally intensive, grid-based representations offer an effective approach to alleviate these challenges. For query, given the Lidar features $B_L \in \mathbb{R}^{C \times H_L \times W_L}$, where $C$, $H_L$, and $W_L$ represent the channel, height, and width of the BEV features, respectively. Then, we fed the BEV feature $B_L$ into a multilayer perceptron (MLP) to obtain Lidar query $Q_L$, which is associated with the initial query features for each Gaussian function ($\mathcal{G}_L \sim Q_L$).

Each voxel in Lidar BEV grid is centered at a 3D location $\boldsymbol{\mu} \in \mathbb{R}^3$, and a Gaussian is initialized at $\boldsymbol{\mu}$ with learnable scale $s$ and rotation $r$. For a Gaussian ($\mathcal{G}_L \sim Q_L$) associated with LiDAR feature $Q_L \in \mathbb{R}^3$, the feature response at a 3D point $\mathbf{p} = (x, y, z)$ within its support is computed as:

$$g_L(\mathbf{p}; \boldsymbol{\mu}, \mathbf{s}, \mathbf{r}) = \exp\big(-\frac{1}{2}(\mathbf{p} - \boldsymbol{\mu})^T \boldsymbol{\Sigma}^{-1}(\mathbf{p} - \boldsymbol{\mu})\big)q_L, \tag{2}$$

### 3.3 Gaussian Encoder

The 3D Gaussian distribution effectively represents the scene, and we have designed a Gaussian encoder to optimize both the properties and query features. This encoder includes a deformable attention with Gaussian module and a Gaussian Updating module. The Gaussian encoder is stacked multiple times to update the Gaussian properties in an iterative refinement paradigm. Additionally, to better fuse multi-modal information, we employ a shared Gaussian encoder to simultaneously process the Gaussian distributions $\mathcal{G}_c$ and $\mathcal{G}_L$, as the two modalities are ultimately intended to converge towards similar Gaussian distributions. Specifically, we merge $\mathcal{G}_c$ and $\mathcal{G}_L$ into the batch dimension.

**Deformable Attention with Gaussian.** As shown in Fig. 2(b), after obtaining the Gaussian distribution sets and corresponding query features ($\mathcal{G}_c \sim Q_c, \mathcal{G}_L \sim Q_L$, we first encode $\mathcal{G}_i \sim Q_i, i = c, L$, into a new query $\hat{Q}$. Specifically, to capture the position and geometric information of the Gaussian functions within the feature maps, we employ an MLP to encode the Gaussian properties. The new query $\hat{Q}$ is then obtained by adding the encoded properties $P_q$ to the original query $Q_i$:

$$\hat{Q}_i = \text{MLP}(\mathcal{G}) + Q_i, i = c, L, \tag{3}$$

where the MLP as the position embedding (PE) transforms the dimensions from $\mathbb{R}^{D_g}$ to $\mathbb{R}^C$. We obtain sets $\mathcal{G}_i \sim \hat{Q}_i$.

Furthermore, we adopt deformable attention with Gaussian (Zhu et al., 2020; H et al., 2024) to extract features, as shown in Fig.3. Vanilla deformable attention (Fig.3(b)) initializes sampling locations with an approximately "square/kernel-like" region and learns offsets to cover the regions of interest. However, this initial distribution lacks inherent geometric priors about object shape. In contrast, our deformable attention with Gaussian (Fig.3(a)) directly inherits and leverages the shape properties of Gaussians: by projecting the 3D Gaussian distributions onto the BEV feature map (projection in Figure 2b), we obtain a prior sampling distribution that encodes orientation, scale, and covariance structure. In other words, the initial sampling points are not uniformly spaced on a grid, but

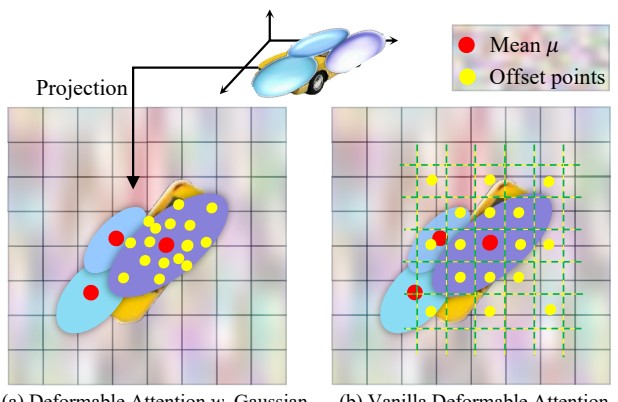

(a) Deformable Attention *w.* Gaussian  (b) Vanilla Deformable Attention

Figure 3: Comparison of the vanilla deformable attention (Zhu et al., 2020) and our proposed deformable attention with Gaussian.

instead follow a Gaussian distribution aligned with the underlying object geometry—such as aspect ratio, orientation, and spatial uncertainty. This Gaussian prior enables better alignment of cross-modal features to the "likely object extent," thereby enhancing fusion effectiveness—a capability absent in conventional square-shaped initialization.

Evidently, the properties of the Gaussian functions effectively describe the shape of the potential objects or regions. For each Gaussian function $g$, we calculate a set of offsets $\Delta\boldsymbol{\mu} = (\Delta x, \Delta y, \Delta z)$ based on the covariance matrix. These offsets, combined with the mean $\boldsymbol{\mu}$, yield the corresponding reference points $\boldsymbol{\mu} + \Delta\boldsymbol{\mu}$. We then project the 3D reference points onto the BEV feature map, where each Gaussian query $q_i \in Q_i$ is updated through deformable attention, expressed as:

$$\text{DeformAtt}(q_i, B_i) = \sum_{k=1}^{K} A_k \cdot W_k B_i(\boldsymbol{\mu} + \Delta\boldsymbol{\mu}), \tag{4}$$

where $B_i$ is the camera BEV feature $B_c$ or Lidar BEV feature $B_L$, $W_k$ is the weights obtained by linear layer, $A_k$ is the attention weights and $A_k \in [0, 1]$, $K$ is the number of sampling points, $B(\boldsymbol{\mu} + \Delta\boldsymbol{\mu})$ are the sampled features.

**Gaussian Updating.** To update the Gaussian properties, we propose an iterative optimization strategy of predicting offsets instead of predicting a set of new Gaussian distributions as adopted in GaussianFormer (H et al., 2024). In Lidar and camera fusion perception, incrementally updating the Gaussian parameters allows for better handling of the discrepancies caused by different modalities when perceiving the same object. This approach is particularly effective in handling fusion uncertainties caused by such as signal attenuation, depth prediction uncertainties, or multi-modal signal discrepancies, as demonstrated by the ablation experiments. Incremental updates across layers allow the model to gradually reduce the disparity between modalities, improving fusion accuracy. Specifically, by predicting the offsets $\Delta\boldsymbol{\mu}$, $\Delta\mathbf{s}$, and $\Delta\mathbf{r}$ for the Gaussian mean $\boldsymbol{\mu}$, scale $\mathbf{s}$, and rotation $\mathbf{r}$ using an MLP, we refine the Gaussian distribution without having to predict a completely new set of properties. The updated Gaussian $\hat{\mathcal{G}}_i$ as follows:

$$\hat{\mathcal{G}}_i = \text{MLP}(\hat{Q}) + \mathcal{G}_i = (\Delta\boldsymbol{\mu} + \boldsymbol{\mu}, \Delta\mathbf{s} + \mathbf{s}, \Delta\mathbf{r} + \mathbf{r}). \tag{5}$$

## 3.4 MULTI-SENSOR FUSION

By the Gaussian Encoder model, we obtain the multimodal 3D Gaussian representations $\hat{\mathcal{G}}_c$ and $\hat{\mathcal{G}}_L$ within the shared 3D space, respectively. We merge $\hat{\mathcal{G}}_c$ and $\hat{\mathcal{G}}_L$ into a unified set $\hat{\mathcal{G}}$, and we can easily fuse them. Although the 3D Gaussian distributions can effectively represent the scene, to handle the irregular distribution of Gaussian points, we need to voxelize these Gaussian distributions to achieve task-independent 3D perception.

Specifically, given the unified Gaussian sets $\hat{\mathcal{G}} \sim \hat{Q}$, we divide the Gaussian space into a voxel grid $H \times W$. For a non-empty voxel $V$ that contains $M$ Gaussian means $\boldsymbol{\mu}$, the Gaussian set for that voxel is $V = \{(\hat{g}_1, \hat{g}_2, \cdots, \hat{g}_M) \sim (\hat{q}_1, \hat{q}_2, \cdots, \hat{q}_M)\}$. To ensure real-time performance and the receptive field of the voxel, we use MeanVFE (Zhou & Tuzel, 2018) to downsample the Gaussian within the voxel, as illustrated in Fig. 2(c). After Gaussian pooling, each voxel contains one Gaussian distribution $(\hat{g} \sim \hat{q})$:

$$\hat{g} = \frac{1}{M}[\sum \boldsymbol{\mu}_m, \sum \mathbf{s}_m, \sum \mathbf{r}_m], \hat{q} = \frac{1}{M}\sum \hat{q}_m. \tag{6}$$

Furthermore, the Gaussian mixture model (Kerbl et al., 2023) can naturally aggregate multiple Gaussian distributions into a finer-grained distribution, unifying multi-modal Gaussian representations and elegantly capturing the complexity of autonomous driving scenes. Thus, if the total number of Gaussian distributions covering point $\mathbf{p}$ in the entire scene is $J$, the feature $f(\mathbf{p})$ at point $\mathbf{p}$ is composed of the cumulative contributions of each individual Gaussian:

$$f(\mathbf{p}) = \sum_{i=1}^{J} \hat{g}_i(\mathbf{p}; \boldsymbol{\mu}, \mathbf{s}, \mathbf{r})\hat{q}_i. \tag{7}$$

Since each voxel may be associated with multiple 3D Gaussian distributions, following the strategy in (H et al., 2024), we calculate the neighborhood radius based on the scale property of each Gaussian. The indices of the Gaussians and the voxels within their neighborhood are paired as tuples

and appended to a list. This list is then sorted by voxel indices, determining which 3D Gaussians each voxel should focus on. Furthermore, for each voxel using 7, we can get the fused feature $B_\text{F}$. Finally, a simple convolutional network is used to further optimize $B_\text{F}$.

### 3.5 PERCEPTION TASK SETUP

Without loss of generality, we follow BEVFusion (Liu et al., 2023b), GaussianFusion can be applied to most 3D perception tasks based on $B_\text{F}$. We evaluate the performance of GaussianFusion on 3D object detection and 3D semantic occupancy prediction tasks. We adopt the same Transformer-based detection head as BEVFusion (Bai et al., 2022; Liu et al., 2023b) and the occupancy head consistent with BEVDet (Huang & Huang, 2022a). We also constructed a camera-only version, GaussianFusion-C, containing only the camera branch shown in Fig. 2.

## 4 EXPERIMENTS

### 4.1 DATASET

The nuScenes dataset (Caesar et al., 2020) provides annotation data for tasks such as semantic segmentation, object detection, and 3D occupancy (Occ) prediction. It is a large-scale multimodal dataset officially split into 700/150/150 scenes for training, validation, and testing, respectively. Each scene includes annotated Lidar point cloud data captured by a 32-beam scanner, along with 6 perspective camera views, offering comprehensive 360-degree coverage at each timestamp. We evaluate our method on the 3D object detection and Occ. In our task, we down-sample the input camera images to $704 \times 256$ and voxelize the point cloud to 0.075m for detection, following BEV-Fusion (Liu et al., 2023b) and UniTR (Wang et al., 2023a). For 3D object detection, the perception range of the point cloud is set to $[-51.2m, 51.2m]$ along the $X$ and $Y$ axes, and $[-5m, 3m]$ along the $Z$ axis. For 3D Occ, we evaluate within the region of $[-50m, 50m] \times [-50m, 50m]$ around the ego vehicle, following (Liu et al., 2023b; Wang et al., 2023a; Wei et al., 2023).

### 4.2 IMPLEMENTATION DETAILS

We adopt VoxelNet (Zhou & Tuzel, 2018) and Swin-T (Liu et al., 2021) as the Lidar and camera backbone to extract the multimodal features following BEVFusion (Liu et al., 2023b). The dimensions of Lidar features, image features, and 3D Gaussian query features are all set to 128. Depth of image $D_c$=41. Image feature $F_{c,i}$ dimensions: $8 \times 22 \times 6$. We set Gaussian Encoder blocks to 4, see the Appendix for experiments. The BEV size $H \times W$ is set to $200 \times 200$.

During training, we follow BEVFusion to adapt the aligned multimodal data augmentation strategy and the class-balanced sampling strategy from CBGS (Zhu et al., 2019). GaussianFusion is trained on 8 NVIDIA A800 GPUs. We use AdamW (Loshchilov & Hutter, 2017) optimizer with a weight decay 0.01. We adopt the one-cycle learning rate policy (Smith, 2017) with a maximum learning rate of $2e^{-4}$. Both BEV object detection and 3D semantic occupancy prediction are trained for 20 epochs, following the same settings as BEVFusion and GaussianFormer (H et al., 2024), respectively.

### 4.3 3D OBJECT DETECTION

**Setting.** We utilize the official evaluation metric nuScenes Detection Score (NDS) and mean Average Precision (mAP) for 3D detection.

**Results.** To highlight the effect of Gaussian representation, we only compare the BEV-based method. As shown in Table 2, GaussianFusion achieves SOTA results compared to previous discrete BEV representation multimodal fusion methods(Liu et al., 2023b; Ge et al., 2023; Wang et al., 2023a; Hu et al., 2023b) on nuScenes dataset, achieving 74.0 NDS and 71.7 mAP on the val split. Specifically, compared with BEVFusion (Liu et al., 2023b), our GaussianFusion achieves +2.6 NDS and +3.2 mAP on val split by exploring a more natural continuous Gaussian cross-modal complementary fusion. In addition, compared with recent SOTA fusion works, such as UniTR (Wang et al., 2023a), EA-LSS (Hu et al., 2023b), and FusionFormer-S (Hu et al., 2023a), GaussianFusion shows superior performance, outperforming them by 1.2, 0.5, and 1.7 respectively in mAP.

Table 2: Comparisons with state-of-the-art 3D object detection methods on nuScenes dataset. C denote Camera, L denote Lidar. All methods construct BEV-based feature maps instead of object-centric fusion based on proposals, which means these methods can also be naturally used for semantic tasks. UniTR uses a unified backbone for both the camera and Lidar.

| Methods | Modality | Resolution | Backbone | | validation set | | test set | |
| | | | Camera | Lidar | NDS | mAP | NDS | mAP |
|---|---|---|---|---|---|---|---|---|
| BEVFormer (Li et al., 2022b) | C | 1600×900 | ResNet-101 | - | 51.7 | 41.6 | 56.9 | 48.1 |
| PETRv2 (Liu et al., 2023a) | C | 1600×640 | VoV-99 | - | - | - | 59.1 | 50.8 |
| FB-BEV (Li et al., 2023) | C | 1600×640 | VoV-99 | - | - | - | 62.4 | 53.7 |
| AutoAlignV2 (Chen et al., 2022) | C+L | 1600×640 | CSPNet | VoxelNet | 71.2 | 67.1 | 72.4 | 68.4 |
| BEVFusion(M) (Liu et al., 2023b) | C+L | 704×256 | Swin-T | VoxelNet | 71.4 | 68.5 | 72.9 | 70.2 |
| MetaBEV (Ge et al., 2023) | C+L | 704×256 | Swin-T | VoxelNet | 71.5 | 68.0 | - | - |
| MSMDFusion (Jiao et al., 2023) | C+L | 800×448 | ResNet-50 | VoxelNet | 72.1 | 69.3 | 74.0 | 71.5 |
| FusionFormer-S (Hu et al., 2023a) | C+L | 1600×640 | VoV-99 | VoxelNet | 73.2 | 70.0 | - | - |
| EA-LSS (Hu et al., 2023b) | C+L | 704×256 | Swin-T | VoxelNet | 73.1 | 71.2 | 74.4 | 72.2 |
| UniTR (Wang et al., 2023a) | C+L | 704×256 | - | - | 73.3 | 70.5 | 74.5 | 70.9 |
| **GaussianFusion(Ours)** | C+L | 704×256 | Swin-T | VoxelNet | **74.0** | **71.7** | **74.9** | **72.4** |

Additionally, we provide a comparison with other open-source state-of-the-art (SOTA) methods in inference latency and performance accuracy in Table 3. Benefiting from the unified architecture, it achieves an excellent performance of 71.7 mAP while maintaining lower inference latency (132 ms) and memory consumption (4271 MB) compared to BEVFusion.

Table 3: Latency and performance on nuScenes *val.* set.

| Method | Latency ↓ | Memory ↓ | NDS ↑ | mAP ↑ |
|---|---|---|---|---|
| BEVFusion | 156 ms | 5140 M | 71.4 | 68.5 |
| GaussianFusion | 132 ms | 4271 M | **74.0** | **71.7** |

Here, we design a simple temporal extension, termed GaussianFusion-T: historical Gaussian representations are warped to the current timestamp and then fused via Equations (5) and (6). In Equation (5), the mean $\mu_m$, scale $s_m$, rotation $r_m$, and query $q^m$ encompass not only the multi-modal (image and Li-DAR) Gaussians and queries, but also those from history frames. Experimental results show that, compared to BEV-

Table 4: Comparison with temporal methods.

| Method | NDS ↑ | mAP ↑ |
|---|---|---|
| BEVFusion4D (Liu et al., 2023b) | 73.5 | 72.0 |
| FusionFormer (Hu et al., 2023a) | 74.1 | 71.4 |
| SparseLIF-T (Zhang et al., 2024a) | 77.5 | 74.7 |
| GaussianFusion-T | **77.6** | **75.0** |

Fusion4D (Liu et al., 2023b), our temporal variant GaussianFusion-T achieves significant improvements. Moreover, even without sophisticated temporal modeling, GaussianFusion-T achieves competitive NDS against advanced temporal fusion methods such as SparseLIF-T (Zhang et al., 2024a).

Table 5: Semantic scene completion results on nuScenes (Wei et al., 2023; Caesar et al., 2020) val set. † represents trained on nuScenes. For Camera-only and C+L, the top performance is indicated in **bold black** and **bold blue**, respectively.

| Method | Modality | IoU | mIoU | barrier | bicycle | bus | car | const. veh. | motorcycle | pedestrian | traffic cone | trailer | truck | drive. suf. | other flat | sidewalk | terrain | manmade | vegetation |
|---|---|---|---|---|---|---|---|---|---|---|---|---|---|---|---|---|---|---|---|
| BEVFormer (Li et al., 2022b) | C | 30.50 | 16.75 | 14.22 | 6.58 | 23.46 | 28.28 | 8.66 | 10.77 | 6.64 | 4.05 | 11.20 | 17.78 | 37.28 | 18.00 | 22.88 | 22.17 | 13.80 | 22.21 |
| TPVFormer (H et al., 2023) | C | 30.86 | 17.10 | 15.96 | 5.31 | 23.86 | 27.32 | 9.79 | 8.74 | 7.09 | 5.20 | 10.97 | 19.22 | 38.87 | 21.25 | 24.26 | 23.15 | 11.73 | 20.81 |
| FB-Occ(1f) (Li et al., 2023) | C | 31.55 | 20.17 | 20.31 | **12.29** | 26.33 | 31.07 | **10.78** | **15.95** | 13.31 | 11.14 | 13.24 | 22.13 | 39.56 | 22.26 | 25.14 | 23.59 | 13.92 | 21.64 |
| SurroundOcc (Wei et al., 2023) | C | 31.49 | 20.30 | 20.59 | 11.68 | 28.06 | 30.86 | 10.70 | 15.14 | 14.09 | 12.06 | **14.38** | 22.26 | 37.29 | **23.70** | 24.49 | 22.77 | **14.89** | 21.86 |
| GaussianFormer (H et al., 2024) | C | 29.83 | 19.10 | 19.52 | 11.26 | 26.11 | 29.78 | 10.47 | 13.83 | 12.58 | 8.67 | 12.74 | 21.57 | 39.63 | 23.28 | 24.46 | 22.99 | 9.59 | 19.12 |
| **GaussianFusion-C** | C | **32.48** | **20.65** | **21.09** | 10.95 | **29.01** | **31.65** | 10.03 | 15.64 | **14.31** | **12.56** | 13.82 | **23.19** | **40.06** | 22.49 | **25.80** | **23.49** | 14.36 | **22.14** |
| BEVFusion† (Liu et al., 2023b) | C+L | 39.11 | 24.65 | 23.78 | 12.29 | 30.67 | 34.95 | 14.62 | 17.23 | 20.76 | 15.75 | 19.83 | 26.3 | 40.01 | 23.34 | 25.47 | 26.41 | 25.15 | 37.92 |
| M-CONet (Wang et al., 2023c) | C+L | 39.20 | 24.70 | 24.80 | 13.00 | 31.60 | 34.80 | 14.60 | 18.00 | 20.00 | 14.70 | 20.00 | 26.60 | 39.20 | 22.80 | 26.10 | 26.00 | 26.00 | 37.10 |
| CO-Occ (Pan et al., 2024) | C+L | 41.10 | 27.10 | 28.10 | 16.10 | 34.00 | 37.20 | 17.00 | 21.60 | 20.80 | 15.90 | 21.90 | 28.70 | 42.30 | 25.40 | 29.10 | **28.60** | 28.20 | 38.00 |
| OccFusion (Ming et al., 2024) | C+L | 43.53 | 27.55 | 25.15 | **19.87** | 34.75 | 36.21 | **20.03** | 23.11 | 25.25 | 17.50 | 22.70 | **30.06** | 39.47 | 23.26 | 25.68 | 27.57 | 29.54 | **40.60** |
| **GaussianFusion** | C+L | **44.75** | **28.65** | **28.92** | 18.31 | **34.87** | **37.43** | 19.45 | **23.53** | **26.71** | **17.96** | **23.38** | 29.89 | **43.32** | **25.96** | **30.51** | 28.35 | **30.32** | 39.67 |

**Waymo Open Dataset Result.** We further conduct experiments on the Waymo Open Dataset (Sun et al., 2020) to evaluate the generalization capability of our approach. GaussianFusion adopts the same heatmap-based detection head as BEVFusion for a fair comparison. Table 6 shows that the Gaussian representation outperforms the BEV representation, achieving higher detection accuracy, which demonstrates its greater potential for 3D perception.

Table 6: Waymo Open Dataset Result.

| Model | mAP-L1 | mAPH-L1 | mAP-L2 | mAPH-L2 |
|---|---|---|---|---|
| BEVFusion (Liu et al., 2023b) | 82.72 | 81.35 | 77.65 | 76.33 |
| GaussianFusion(ours) | 86.23 | 84.65 | 81.47 | 80.75 |

## 4.4 3D SEMANTIC OCCUPANCY PREDICTION

**Setting.** We report the Intersection-over-Union (IoU) of occupied voxels as the evaluation metric of the class-agnostic scene completion task and the mIoU of all semantic classes for the Occ task following SurroundOcc (Wei et al., 2023).

**Results.** As shown in Table 5, our GaussianFusion achieves SOTA performance at 28.65 mIoU among all single-frame models. GaussianFusion outperforms the multi-modal SOTA method OccFusion (Ming et al., 2024), which is based on multi-scale voxel fusion, by +1.11 mIoU and significantly surpasses camera-only methods (H et al., 2023;

Table 7: Comprehensive comparison with Gaussian-Former on nuScenes val set.

| Method | Num. Gaussians | mIoU ↑ | Latency ↓ |
|---|---|---|---|
| GaussianFormer | 140,000 | 19.10 | 475 ms |
| GaussianFusion-C | 43,296 | **20.65** | **105 ms** |

Wei et al., 2023; Li et al., 2023). More importantly, benefiting from our proposed Gaussian initialization strategy and iterative update mechanism, GaussianFusion-C achieves a 1.55 mIoU improvement and nearly 4.5× computational efficiency compared to GaussianFormer, while using only 30% of the Gaussians, as shown in Table 7. GaussianFormer randomly initializes a set of Gaussians in 3D space and predicts new Gaussian parameters for these Gaussians in an update. Extensive experiments demonstrate the effectiveness of our 3D Gaussian representation across multiple tasks, including both object-centric and dense semantic perception.

Table 8: Ablation of Gaussian initialization strategy.

| Gaussian Initialization | NDS | mAP |
|---|---|---|
| Random Initialization | 71.2 | 68.3 |
| Backward Projection | 72.4 | 70.0 |
| Lidar Projection | 73.6 | 71.1 |
| **Forward Projection** | **74.0** | **71.7** |

Table 9: Ablation of the proposed Gaussian Encoder. DA.G means Deformable Attention with Gaussian.

| Share | Separate | DA.G | PE | Offset | NDS | mAP |
|---|---|---|---|---|---|---|
| ✓ | | ✓ | ✓ | ✓ | **74.0** | **71.7** |
| ✓ | | | ✓ | ✓ | 73.6 | 71.1 |
| | ✓ | ✓ | ✓ | ✓ | 73.4 | 71.0 |
| ✓ | | ✓ | | ✓ | 73.6 | 71.2 |
| ✓ | | ✓ | ✓ | | 73.2 | 70.8 |

## 4.5 ABLATION STUDIES

**Effect of Gaussian Initialization.** Table 8 provides a detailed analysis of the performance impact of different Gaussian initialization strategies, including the classic random initialization, our proposed forward projection strategy, the backward projection BEVFormer-based strategy(Li et al., 2023; 2022b), and the strategy of projecting Lidar points onto the image. The initialization details and corresponding Gaussian encoder for the latter two projection strategies are in Appendix. The forward projection and Lidar projection strategies show comparable performance (74.0 NDS $v.s$ 73.6 NDS), both outperforming the backward projection method (72.4 NDS). Notably, the forward projection strategy brings a significant improvement of +2.8 NDS over random initialization.

**Effect of Gaussian Encoder.** In Table 9, we first compare the shared and separate Gaussian Encoders. We find that the shared Gaussian Encoder provides a performance improvement of +0.7

mAP. We attribute this to the unified Gaussian space, which helps the model learn uncertain cross-modal complementary features. And the sharing strategy makes the model leaner. We then conduct an ablation study on the deformable attention module. Results show that deformable attention with Gaussian priors outperforms the vanilla variant by +0.4 NDS, demonstrating that the shape prior encoded by Gaussians facilitates model convergence and enhances detection accuracy. For the deformable attention-based query updating module, the results show that encoding the Gaussian properties as PE into the query leads to a gain of +0.5 mAP. For the Gaussian updating module, predicting the offsets of the properties, rather than the properties themselves, improves multimodal fusion perception by +0.9 mAP. This validates the effectiveness of the Gaussian encoder.

## 4.6 VISUALIZATIONS

As shown in Fig. 4, in BEV object detection, compared to previous BEV-based SOTA methods like UniTR (Wang et al., 2023a) and BEVFusion(Liu et al., 2023b), GaussianFusion achieve higher accuracy for distant or small objects (white marks). Furthermore, UniTR exhibits significant object yaw errors (yellow marks). For Occ, GaussianFusion-C produces sharper object boundaries (red marks) and better class separation (yellow marks) compared to GaussianFormer. See Appendix for more visualizations.

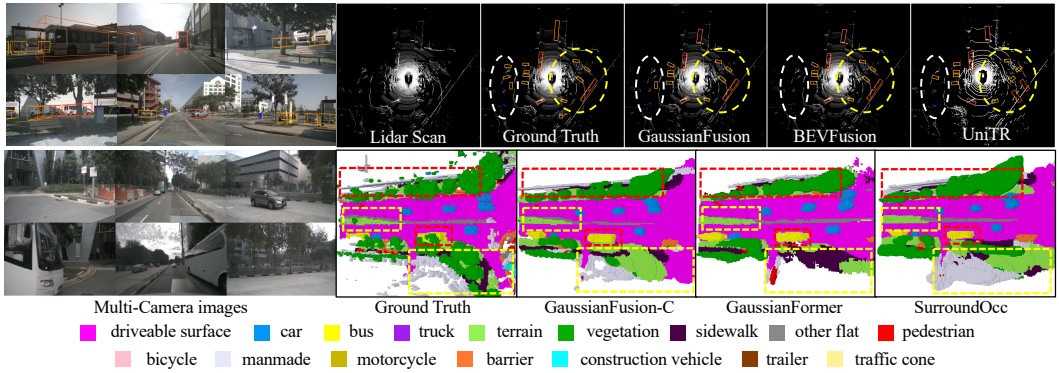

Figure 4: Qualitative results on object detection and 3D semantic occupancy prediction.

## 4.7 LIMITATIONS

Several approaches—covering both detection (Wang et al., 2023b) and Occ (Zhang et al., 2024b)—employ carefully designed temporal fusion modules to enhance performance. While our method naturally extends to multi-frame settings through simple temporal alignment and already achieves performance comparable to such multi-frame methods, this is likely suboptimal. A promising direction for future work is to explore motion-aware Gaussian updates, for instance by predicting velocity-guided offsets, enabling more coherent 4D scene modeling over time.

## 5 CONCLUSION

We present GaussianFusion, a novel multi-modal fusion perception framework grounded in a unified 3D Gaussian representation that seamlessly integrates camera and LiDAR features in a continuous spatial domain, effectively preserving fine-grained scene details. A shared Gaussian encoder is introduced to facilitate adaptive cross-modal interaction and alignment, with Gaussian properties iteratively refined through optimization. To support task-agnostic applications, we design an efficient Gaussian-to-voxel transformation module incorporating Gaussian pooling and aggregation mechanisms. Extensive experiments across multiple 3D perception tasks on the nuScenes dataset validate the effectiveness of GaussianFusion, achieving state-of-the-art performance among task-agnostic baselines. Although it may slightly lag behind certain task-specialized approaches, our work represents a meaningful step toward generalizable and principled multi-modal fusion. We believe that the proposed Gaussian representation paradigm offers a promising direction for future research in multimodal 3D perception.

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
