## A  APPENDIX

### A.1  NETWORK ARCHITECTURE

**Gaussian initialization.** For the Lidar projection strategy, we first voxelize the point cloud and retain **P** points per voxel, padding with zeros if fewer than **P** points are available. Next, the point cloud is projected onto the image, and each point's 3D information is used to initialize the Gaussian mean. Finally, each Gaussian query extracts features from the image via cross-attention, as shown in Fig. 5, to update the 3D Gaussian distribution from the image. Additionally, consistent with BEVFormer (Li et al., 2022b), only the view where the query projection resides is considered, rather than all six views. For the backward projection (Li et al., 2022b; 2023) strategy, we construct the BEV prior distribution, project it back onto the image, and update the Gaussian distribution through deformable cross-attention (as shown in Fig. 5).

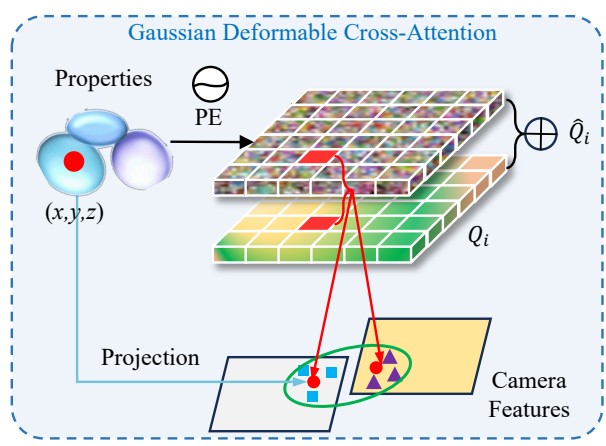

Figure 5: The 2D image feature to 3D Gaussian feature transformation module based on deformable cross-attention (Zhu et al., 2020).

## B  ROBUSTNESS AGAINST SENSOR FAILURE

Sensor failures are likely to occur in practical production. To demonstrate the robustness of our GaussianFusion in the presence of Lidar and camera failures, we follow the same evaluation protocols adopted in UniTR (Wang et al., 2023a) and BEVFusion (Liang et al., 2022; Liu et al., 2023b). We refer the readers to previous papers (Liang et al., 2022) for more details. Tab. 10 and 11 shows that under certain camera or Lidar beam failure scenarios, our GaussianFusion achieves SOTA results compared to previous advanced BEV representation methods, demonstrating the robustness of the Gaussian-based multi-modal fusion paradigm under sensor failure.

Table 10: Evaluating the robustness on degradation conditions. "F" is the front camera, and Stuck means unsynchronized timestamp between LiDAR and cameras.

| Method | Clean | | MissingF | | PreserveF | | Stuck | |
|---|---|---|---|---|---|---|---|---|
| | mAP | NDS | mAP | NDS | mAP | NDS | mAP | NDS |
| BEVFusion (Liu et al., 2023b) | 67.9 | 71 | 65.9 | 70.7 | 65.1 | 69.9 | 66.2 | 70.3 |
| UniTR (Wang et al., 2023a) | 70.5 | 73.3 | 68.5 | 72.4 | 66.5 | 71.2 | 68.1 | 71.8 |
| GaussianFusion (ours) | **71.7** | **74.0** | **70.1** | **73.0** | **67.7** | **71.3** | **69.4** | **72.3** |

Table 11: Lidar beam loss ablation.

| Method | 1-beam | 4-beam | 16-beam | 32-beam |
|---|---|---|---|---|
| BEVFusion (Liu et al., 2023b) | 52.0 | 63.2 | 64.4 | 71.4 |
| UniTR (Wang et al., 2023a) | 59.5 | 68.5 | 72.2 | 73.3 |
| GaussianFusion (ours) | **64.4** | **70.8** | **73.1** | **74.0** |

## C    MULTI-TASK LEARNING

We present the performance evaluation of joint multi-task training, including 3D object detection and BEV map segmentation. It is important to note that our GaussianFusion focuses on multimodal feature fusion rather than multi-task learning (MTL). We follow the MTL training strategy used by BEVFormer (Liu et al., 2023b). Such as the loss ratio between the 3D object detection and BEV map segmentation tasks is set to 1:1 and separate multi-task BEV encoder performance improvement strategy. As shown in Table 12, our GaussianFusion's MTL variant outperforms previous powerful UniTR (Wang et al., 2023a).

Table 12: Joint training. All experiments are conducted on the nuScenes val set.

| Method | MTL | mAP | NDS | mIoU |
|---|---|---|---|---|
| BEVFusion (Liu et al., 2023b) | shared | - | 69.7 | 54.0 |
| UniTR (Wang et al., 2023a) | | 67.6 | 71.4 | 69.5 |
| **GaussianFusion** | | **68.9** | **71.5** | **70.9** |
| BEVFusion (Liu et al., 2023b) | separate | 65.8 | 69.8 | 58.5 |
| UniTR (Wang et al., 2023a) | | 68.2 | 71.6 | 71.2 |
| **GaussianFusion** | | **69.8** | **72.1** | **71.6** |

## D    MORE DISCUSSION AND LIMITATIONS

In object-oriented tasks, we adopt a dense fusion strategy by lifting 2D camera to 3D space, which is inevitably more time-consuming than object-centric fusion. Moreover, current advanced object-centric fusion methods (Vora et al., 2020; Chen et al., 2023; Zhou & T, 2023; Yin et al., 2024; Li et al., 2024; Wang et al., 2024b) enhance multi-modal 3D detection by leveraging 2D detection results from the camera, achieving impressive performance. Although GaussianFusion performs slightly lower than some of these methods, such as (Yin et al., 2024), it is noteworthy that our approach is designed for general-purpose tasks, including 3D object detection and 3D occupancy prediction, etc. We believe that there is potential to design object-oriented models based on 3D Gaussian representations.

## E    MORE VISUALIZATIONS

Fig. 6 presents a visual comparison with the multi-modal fusion method OccFusion. The results show consistent performance between the multi-modal GaussianFusion and the camera-only GaussianFusion-C, demonstrating that GaussianFusion better distinguishes between different object categories (highlighted in red) and achieves a lower prediction error rate (indicated in yellow). This validates the effectiveness of the Gaussian-based multi-modal fusion framework.

Fig. 7 illustrates a failure case, where both our GaussianFusion-C and the prior GaussianFormer miss parts of the drivable road surface, while the BEV-based SurroundOcc produces correct occupancy predictions (as marked in red). A possible explanation for this behavior lies in the inherent reliance of Gaussian representations on learned instance-aware density distributions. Unlike BEV-based methods, which explicitly model occupancy on a dense, regular grid—naturally capturing large-scale structures such as roads—Gaussian representations typically focus on compact, object-centric primitives. As a result, they may under-reconstruct spatially extended but texture-poor regions like road surfaces.

In Fig. 8, we present visualization results of GaussianFusion on BEV 3D object detection on the nuScenes dataset. Numerous examples demonstrate that our method achieves predictions closer to ground truth compared to previous BEV-based SOTA methods like UniTR (Wang et al., 2023a) and BEVFusion (Liu et al., 2023b). Furthermore, considering the mAP metric in Table 2, we observe that the SOTA UniTR shows a significant performance drop on mAP. The IoU calculation takes into account the position, size, and orientation of bounding boxes, which aligns with the first and third rows examples of Fig. 8. In contrast, our GaussianFusion demonstrates robust geometric and semantic modeling, achieving strong performance.

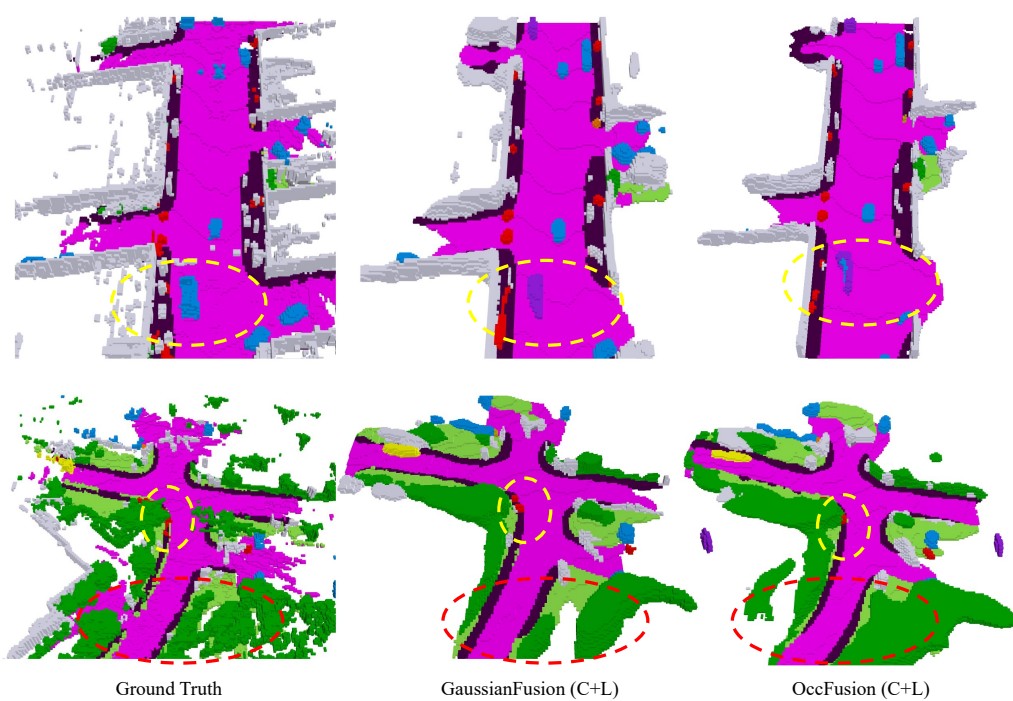

| Ground Truth | GaussianFusion (C+L) | OccFusion (C+L) |

Figure 6: Qualitative comparison with OccFusion.

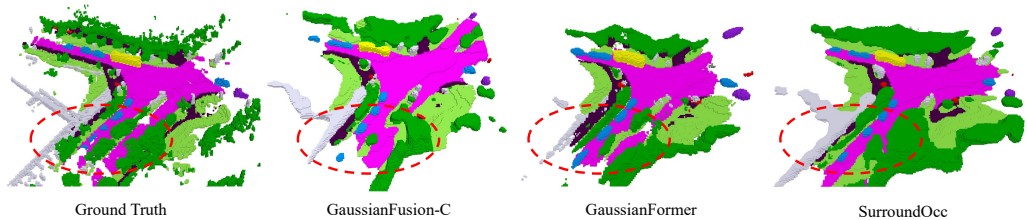

| Ground Truth | GaussianFusion-C | GaussianFormer | SurroundOcc |

Figure 7: Analysis of Failure Cases.

Fig. 8 also provides a qualitative comparison of GaussianFusion-C with GaussianFormer (H et al., 2024) and SurroundOcc (Wei et al., 2023) on 3D semantic occupancy. As highlighted by the (low missed prediction rate) and yellow (better distinguishes between different object classes) markers, our Gaussian representation paradigm and optimized model show clear advantages over previous Gaussian-based or voxel-based methods. Similarly, there are many methods (Lu et al., 2023; Ma et al., 2024a) for optimizing 3D semantic occupancy, which we will not go into detail here.

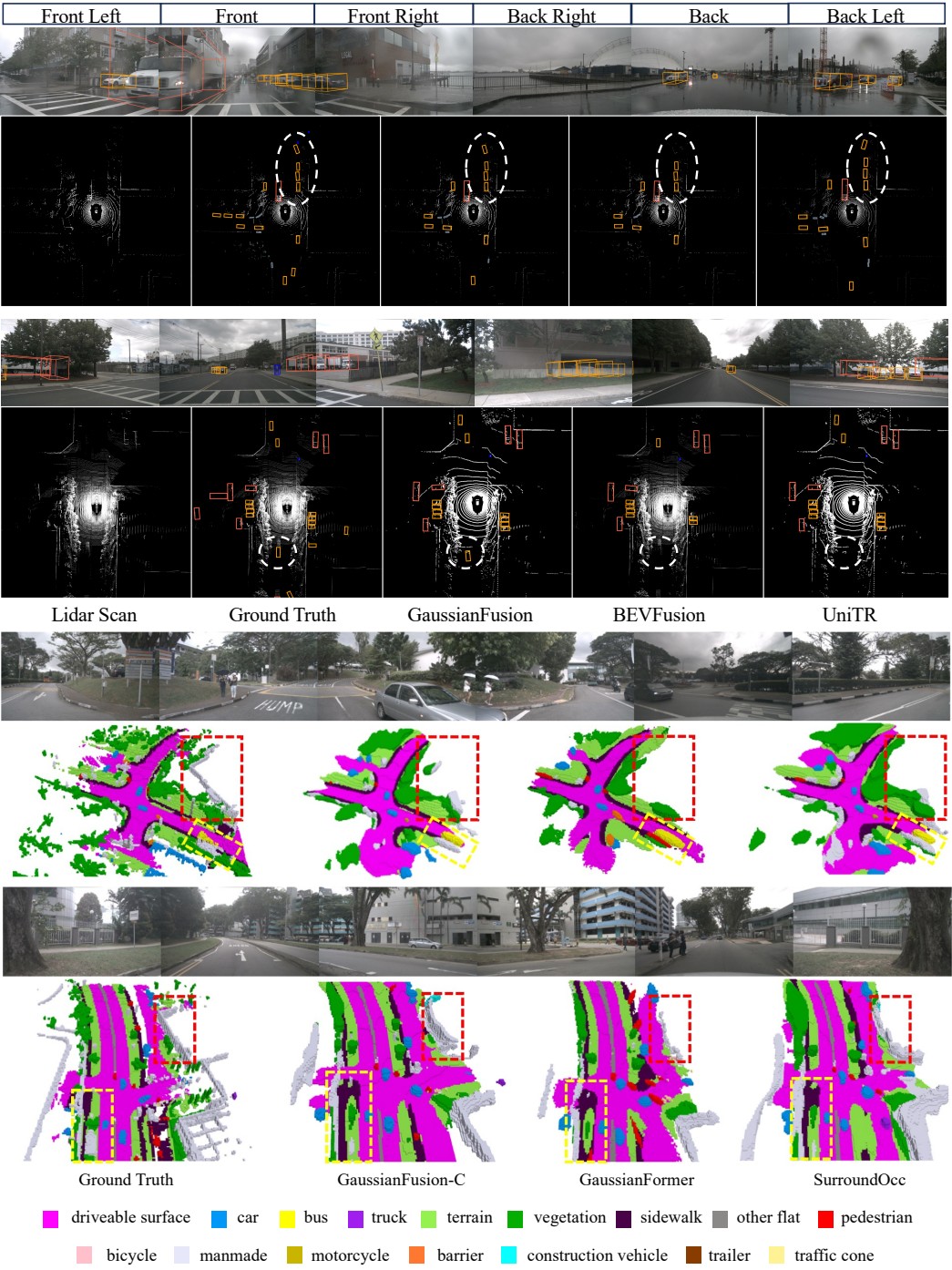

Figure 8: Qualitative comparison BEV object detection and 3D semantic occupancy prediction on nuScenes validation set. Better viewed when zoomed in.