# OpenReview forum: "GaussianFusion: Unified 3D Gaussian Representation for Multi-Modal Fusion Perception"
_ICLR.cc/2026/Conference — ICLR 2026 Poster_

### Official Review · Reviewer_mxcM · 2025-10-30

**Soundness:** 3
**Presentation:** 2
**Contribution:** 2
**Rating:** 2
**Confidence:** 5

**Summary:**

GaussianFusion presents a unified 3D Gaussian-based framework for multimodal fusion, replacing discrete BEV with a continuous shared space. It aligns camera and LiDAR features via a shared encoder with deformable attention and iterative updates. A Gaussian-to-voxel module enables task-agnostic perception. On nuScenes, it achieves state-of-the-art results in detection and occupancy tasks with lower latency and memory.

**Strengths:**

1. Unified continuous 3D Gaussian representation eliminates early discretization and preserves fine geometric/semantic detail.

2. Forward-projection initialization produces accurate, optimization-friendly Gaussians without random guesswork.

3. Shared Gaussian encoder with deformable attention enables natural cross-modal alignment and iterative refinement.

4. Gaussian-to-voxel pooling delivers task-agnostic features for both detection and dense occupancy, beating BEV baselines while using less memory and latency.

**Weaknesses:**

1. It lags behind sparse-detection specialists that optimize only for 3D object recall.

2. No temporal modeling is incorporated, so multi-frame cues remain unused.

3. Gaussian-to-voxel conversion still imposes a fixed voxel grid, re-introducing minor discretization error.

4. The shared encoder assumes synchronized full-modal inputs; robustness to sensor dropout is not evaluated.

**Questions:**

1. Have experiments been conducted on datasets such as nuPlan, NavSim, Waymo, or CARLA?

2. Where is the real-time performance analysis that the autonomous-driving community cares about?

3. What are the advantages of Gaussian-to-Voxel Pooling over BEV Pooling and Voxel R-CNN's Voxel ROI Pooling, and why weren't corresponding ablation experiments conducted?

4. In Figure 2, the schematic diagrams in both the overall part (a) and part (b) do not clearly illustrate the correspondence with the inputs, leading to potential ambiguity and necessitating revision.

5. What exactly is the "&" operation in Figure 2? It is not explicitly defined in the paper and needs to be clarified by the authors.

6. The authors need to clarify whether the two tasks are trained separately or handled simultaneously by a single model; Figure 2 suggests joint inference, but the presentation is ambiguous.

7. This work proposes to encode features by combining BEV and Gaussian representations, yet the detection head is not described in detail. Moreover, the paper does not explore whether alternative heads could be plugged in to verify the generality of the proposed encoding.

---

> ### Author Response · Authors · 2025-11-20
>
> Weaknesses:
> 1.	It lags behind sparse-detection specialists that optimize only for 3D object recall.
>
> We believe there may be a slight misunderstanding. We clarify that our core contribution is a continuous Gaussian-based scene representation to mitigate quantization artifacts from voxelization—rather than designing a novel detection head. To isolate its impact, we use the same detection/occupancy heads as BEVFusion, UniTR, and GaussianFormer for fair comparison.
>
> Importantly, our framework supports advanced query-based heads (e.g., CMT-style): image and LiDAR Gaussians serve as 3D-encoded KV inputs, while queries are decoded for box prediction. With this upgrade, our method achieves competitive NDS and +1.7 mAP over SparseLIF—the state-of-the-art sparse detector optimized for 3D object recall.
>
> detection head	NDS	mAP
>
> SparseFusion	73.1	71.0
>
> GaussianFusion（heatmap-based）	74.0	71.7
>
> IS-FUSION	74.0	72.8
>
> SparseLIF	74．6	71.2
>
> GaussianFusion（query-based）	74.5	72.9
>
> This shows that our representation not only enables fair comparison, but also supports direct integration with modern detection paradigms—delivering performance on par with specialized sparse methods, while maintaining a continuous, high-fidelity scene model.
>
> 2.	No temporal modeling is incorporated...
>
> As in Q1, our focus is on Gaussian representation as a continuous, generalizable scene model. We extend it temporally with GaussianFusion-T: historical Gaussians are warped and fused using our formulation (Eqs. 5–6), inspired by BEVDet4D/BEVFusion4D.
> Results show clear gains over BEVFusion4D and competitive NDS versus SparseLIF-T—despite not specializing in sparse detection. This demonstrates that Gaussian representation naturally supports temporal modeling, reinforcing its potential as a scalable, general-purpose scene representation.
>
> Method	Modality	NDS	mAP
>
> BEVFusion4D	C+L	73.5	72.0
>
> FusionFormer	C+L	74.1	71.4
>
> SparseLIF-T	C+L	77.5	74.7
>
> GaussianFusion-T	C+L	77.6	75.0
>
>
>
>
> 3.Gaussian-to-voxel conversion still imposes a fixed voxel grid
>
> Our Gaussian representation supports query-based detection (e.g., CMT-style): image and LiDAR Gaussians serve as transformer KV features, with detection queries processed via a Transformer decoder. This avoids voxelization entirely.
>
> Results show improved accuracy over heatmap-based head, confirming that quantization in BEV voxelization harms performance, while continuous Gaussian modeling enhances perception quality.
>
> Detection head	NDS	mAP
>
> GaussianFusion（heatmap-based）	74.0	71.7
>
> GaussianFusion（query-based）	74.5(+0.5)	72.9(+1.2)
>
>
>
> 4. Sensor dropout experiments.
>
> This is a valuable suggestion. To further evaluate the robustness of our method under real-world sensing conditions, we have conducted additional experiments:
>
> Evaluating the robustness on degradation conditions. “F” is the front camera, and Stuck means unsynchronized timestamp between LiDAR and cameras.
>
> Method	Clean	MissingF 	PreserveF 	Stuck
> 	mAP	NDS	mAP	NDS	mAP	NDS	mAP	NDS
>
> BEVFusion	67.9	71	65.9	70.7	65.1	69.9	66.2	70.3
>
> UniTR	70.5	73.3	68.5	72.4	66.5	71.2	68.1	71.8
>
> GaussianFusion	71.7	74.0	70.1	73.0	67.7	71.3	69.4	72.3
>
>
>
> Questions:
>
> 1. nuPlan, NavSim, Waymo, or CARLA experiments
>
> We thank the reviewer for the suggestion. While nuPlan, NavSim, and CARLA focus on end-to-end planning, our work targets perception-level 3D scene understanding. To validate our Gaussian representation, we add experiments on the Waymo Open Dataset—a standard 3D perception benchmark. Results show consistent improvements over BEVFusion across metrics and difficulty levels, demonstrating GaussianFusion's effectiveness in diverse, real-world scenarios.
>
> Model	mAP-L1	mAPH-L1	mAP-L2	mAPH-L2
>
> BEVFusion 82.72	81.35	77.65	76.33
>
> GaussianFusion（ours）	86.23	84.65	81.47	80.75
>
>
> 2.  Real-time performance..
>
> We have analyzed inference time in Table 3 of the main paper. As demonstrated throughout the paper, GaussianFusion achieves faster inference speed than BEVFusion and GaussianFormer, with GaussianFusion-C running 4.5× faster than GaussianFormer (Table 5). This efficiency advantage is a key benefit of our approach. In addition, we have conducted a detailed analysis of the computational complexity in Appendix Table 10.

---

> ### Author Response · Authors · 2025-11-20
> **Continue the above reply**
>
> 3. BEV Pooling and  Voxel ROI Pooling
>
> Our paper shows Gaussian representation is continuous, avoiding BEV’s quantization loss and enabling more accurate modeling. Finer BEV grids improve performance but with high cost (Table 1), revealing inherent limitations. In contrast, Voxel-RCNN uses voxel ROI pooling for LiDAR-only detection—focused on single-modality extraction, unlike our multi-modal fusion goal.
>
> 4. Figure 2 schematic diagrams.
>
> We clarify: Section 3 and Figure 2 detail our method. Camera inputs use LSS for depth-guided Gaussian initialization (gc ) and a context network for queries (Qc ). LiDAR inputs initialize gL directly from points (200×200 grid), with QL from VoxelNet. The 3D Gaussian shape captures object extent; projecting them to BEV yields sampling locations in deformable attention, shown as ellipses in Figure 2(b)—guiding feature refinement with spatial priors.
>
> 5. What exactly is the "&" operation in Figure 2?
>
> The symbol “&” denotes “and”, indicating the association between the modality-specific Gaussian g  and its corresponding query Q.
>
> 6. The authors need to clarify whether the two tasks…
>
> We follow the visualization style of BEVFusion and UniTR, and “Task-Specific Perception” implies separate heads. In response, we will clarify in Figure 2 that inference is performed independently per task. Nevertheless, our framework supports joint inference, as demonstrated by the new BEVFusion-style multi-task joint training experiments in the appendix.

---

### Official Review · Reviewer_x14q · 2025-10-30

**Soundness:** 2
**Presentation:** 3
**Contribution:** 2
**Rating:** 4
**Confidence:** 3

**Summary:**

This paper proposes *GaussianFusion*, a multi-modal fusion framework for 3D perception in autonomous driving. The core idea is to replace discrete Bird’s-Eye View (BEV) representations with a unified 3D Gaussian representation to preserve fine-grained geometric and semantic details. Key contributions include:

- A **forward-projection-based Gaussian initialization** strategy for camera and LiDAR data, avoiding random initialization used in prior work (e.g., GaussianFormer).
- A **shared Gaussian encoder** with deformable attention and iterative property updates to align cross-modal features.
- A **Gaussian-to-voxel fusion module** for task-agnostic 3D perception.
- State-of-the-art results on nuScenes for 3D object detection (74.0 NDS) and semantic occupancy prediction (28.65 mIoU), with improved efficiency over GaussianFormer.

**Strengths:**

**Originality**:

- The use of 3D Gaussian representations for multi-modal fusion is novel. While 3D Gaussian Splatting (3DGS) has been explored for reconstruction (Kerbl et al., 2023) and single-modal perception (H et al., 2024), its application to multi-modal fusion is a creative extension.
- The forward-projection initialization addresses a critical limitation of random initialization in GaussianFormer, improving convergence and performance (Table 6).

**Quality**:

- Experiments are thorough, covering both object detection and occupancy prediction. The ablation studies (Tables 6–7) validate design choices, and comparisons with BEVFusion (Liu et al., 2023b) and GaussianFormer (H et al., 2024) are compelling.
- The shared encoder and Gaussian mixture model (Eq. 4–5) elegantly unify cross-modal features.

**Clarity**:

- The paper is well-structured, with clear figures (e.g., Fig. 2) illustrating the pipeline. Mathematical formulations (e.g., covariance matrix derivation in §3.2) are precise.

**Significance**:

- The framework advances multi-modal perception by addressing BEV’s quantization limitations. The 450% speedup over GaussianFormer (Table 5) demonstrates practical value.

**Weaknesses:**

**1. Limited Novelty in Fusion Mechanism**:

- While the Gaussian representation is novel, the fusion mechanism (shared encoder + deformable attention) closely follows prior work (e.g., BEVFusion’s CNN fusion, UniTR’s cross-modal attention). The paper does not sufficiently differentiate its fusion strategy from existing methods. For example, the deformable attention module (Fig. 2b) resembles Zhu et al. (2020) without novel adaptation for Gaussian properties.

**2. Incomplete Comparison with Sparse Detectors**:

- The authors acknowledge that sparse detectors (e.g., Zhang et al., 2024a; Li et al., 2024) outperform their method but dismiss them as “task-specific.” However, recent sparse fusion works (e.g., Wang et al., 2024b; Yin et al., 2024) achieve both high detection accuracy and occupancy prediction capability. A direct comparison with these methods is missing and weakens the claim of “task-agnostic” superiority.

**3. Shallow Technical Details**:

- Critical components lack implementation specifics:
  - The forward-projection initialization for LiDAR (§3.2) is described as “BEV-based” but lacks mathematical formulation (unlike the camera case).
  - The Gaussian mixture aggregation (Eq. 5) is presented as a summation without addressing computational complexity or optimization strategies for large-scale scenes.

**4. Under-Explored Limitations**:

- The computational overhead of Gaussian-to-voxel conversion (§3.4) is glossed over. For example, the “neighborhood radius” calculation for Gaussian-voxel pairing is mentioned but not analyzed (e.g., runtime vs. BEV resolution).
- The method’s reliance on dense LiDAR data (nuScenes) raises questions about its applicability to low-cost, camera-only systems.

**5. Weak Visualization**:

- Qualitative results (Fig. 3) are limited to two scenarios and lack comparisons with key baselines (e.g., OccFusion). The claimed “sharper boundaries” are not quantitatively validated.

**Questions:**

**Questions**:

1. How does the shared encoder resolve modality-specific conflicts (e.g., camera depth uncertainty vs. LiDAR precision)? Is there a mechanism to weight modalities dynamically?
2. The Gaussian updating module (§3.3) predicts offsets (Δµ, Δs, Δr). How stable is this optimization? Does it suffer from gradient vanishing/explosion with multiple encoder layers?
3. Could the framework support temporal fusion (e.g., multi-frame LiDAR/camera data)? The authors mention it as future work but provide no preliminary analysis.

**Suggestions**:

1. **Strengthen the fusion novelty**: Compare the deformable attention mechanism with prior Gaussian-guided attention (e.g., Zuo et al., 2024) and highlight differences.
2. **Expand experiments**:
   - Compare with recent sparse fusion methods (e.g., Wang et al., 2024b; Yin et al., 2024).
   - Report computational metrics (FLOPS, memory) for Gaussian-to-voxel conversion.
3. **Clarify technical details**:
   - Provide equations for LiDAR Gaussian initialization.
   - Discuss how Gaussian scale (**s**) and rotation (**r**) are regularized during training.
4. **Enhance visualization**: Include failure cases (e.g., occluded objects) and per-class IoU curves for occupancy prediction.

---

**Rebuttal Potential**: The authors could improve the rating by:

- Demonstrating superior task-agnostic performance against sparse fusion baselines.
- Providing deeper analysis of computational trade-offs.
- Adding a ablation on the Gaussian mixture aggregation strategy.

Overall, the paper presents a promising direction but requires stronger differentiation from existing fusion paradigms and more rigorous evaluation to meet the acceptance threshold.

---

> ### Author Response · Authors · 2025-11-20
>
> Weaknesses：
> 1. Strengthen the fusion novelty:
>
> Unlike vanilla deformable attention, which starts from a uniform grid (resembling a square kernel) without shape prior, our Deformable Attention with Gaussian leverages the geometric properties of 3D Gaussians to guide sampling. By projecting the Gaussian onto the BEV feature map (Fig. 2b), aligning with the object shape. This shape-aware prior enables more accurate alignment of multi-modal features along potential object boundaries, enhancing fusion accuracy.
>
> Ablation studies (as follows) show our method outperforms vanilla deformable attention. We have further updated the figure illustrations to clarify the difference in main paper.
>
> Method						 NDS  mAP
>
> Vanilla Deformable Attention	        73.6	71.1
>
> Deformable Attention w. Gaussian	74.0	71.7
>
> 2. Incomplete Comparison with Sparse Detectors:
>
> These (do you mean Mv2dfusion and Is-fusion？) are detection-only, query-based sparse methods with object-centric features, not designed for occupancy. Conversely, SparseOcc supports occupancy but not detection. In contrast, task-agnostic BEV-based methods (e.g., BEVFusion, PanoOcc) enable task-specific heads—our comparison basis.
>
> GaussianFusion replaces BEV with geometrically richer 3D Gaussians. Using the same detection head as BEVFusion/ ensures fair comparison. Crucially, it can adopt advanced query-based heads (e.g., CMT-style), achieving comparable NDS/mAP to Mv2dFusion (see below), proving Gaussian superiority over BEV.
>
> Detection head	NDS	mAP
>
> SparseFusion	        73.1	71.0
>
> IS-FUSION		74.0	72.8
>
> Mv2dfusion		74.7 72.8
>
> GaussianFusion(CMT) 74.5 72.9
>
> 3. Shallow Technical Details:
>
> Each voxel in Lidar BEV grid is centered at a 3D location μ∈R3, and a Gaussian is initialized at μ with learnable scale s and rotation r. For a Gaussian gL∼qL associated with LiDAR feature qL∈RC, the feature response at a 3D point p=(x,y,z)within its support is computed as: gL(p;μ,s,r)=exp(−12(p−μ)⊤Σ−1(p−μ))⋅qL
>
> 4.1 FLOPS, memory for Gaussian-to-voxel conversion:
>
> We address these questions collectively. In our Gaussian-to-voxel module, the mean operation ensures at most one Gaussian per voxel, effectively sparsifying the representation and reducing computational load. Under this design, the overall complexity scales as M = (H x W)r, where r is the number of voxels influenced per Gaussian—controlled by a fixed splatting radius (e.g., 1.5m).
>
> As shown in the table below, this allows efficient scaling across BEV resolutions. For large scenes (e.g., 400×400), we can further reduce computation by slightly decreasing the splatting radius, with minimal impact on performance. This spatial sparsity and radius adaptability make our method scalable to large-scale deployments.
>
> BEV Size	Voxel Size Splatting Radius	Total Pairs	FLOPs	Memory	Latency
>
> 100×100	1.008m	2	1.25M	57.5M	0.72 MB		~2ms
>
> 200×200	0.504m	3	13.72M	631M	2.88 MB		~15 ms
>
> 400×400	0.252m	6	351.5M	16.2G	11.52 MB	~320 ms
>
> 400×400	0.252m	5	213.2M	9.8G	11.52 MB	~190 ms
>
> Camera-only systems.
>
> In the main paper, we present a camera-only variant, GaussianFusion-C, for 3D occupancy prediction. As shown in Table 5 of the main paper, it achieves a +1.55 mIoU improvement over the pure vision method GaussianFormer, while running at nearly 4.5× higher computational efficiency and using only 30% of the Gaussians. This demonstrates the superior representational efficiency and effectiveness of our framework even in the Camera-only systems.
>
> Questions: 1.How does the shared encoder resolve modality-specific conflicts…
>
> Our framework achieves robust dynamic fusion by combining deformable attention, iterative refinement, and a shared encoder—enabling automatic "trust allocation" based on geometric consistency:
>
> （1）Dynamic weighting: Deformable attention assigns higher scores to confident, well-aligned Gaussians, lower ones to uncertain inputs, acting as an implicit confidence mechanism.
> （2）Iterative refinement: A multi-layer encoder gradually resolves conflicts.
> （3）Shared optimization: Shared parameters ensure convergence in mutually supported regions.
>
> 2. The Gaussian updating module (§3.3) predicts offsets…
>
> The offset-based Gaussian update is stable in practice. We use residual updates with small initialization and layer-wise normalization.  Empirically, training converges smoothly across all experiments.
>
> 3. Could the framework support temporal fusion…
>
> Dense methods (e.g., BEVDet4D, BEVFusion4D) warp historical BEV features and fuse them via deformable attention or 3D convolutions. In contrast, GaussianFusion-T warps historical 3D Gaussians and fuses them using Eqs. (5)–(6).
> As below, GaussianFusion-T significantly outperforms BEVFusion4D and achieves competitive NDS against SparseLIF-T.
> Further, modeling motion via velocity-aware updates could enable more coherent 4D scene understanding—a promising direction for future work.
>
> Method			NDS	mAP
>
> BEVFusion4D		73.5	72.0
>
> FusionFormer		74.1	71.4
>
> SparseLIF-T		77.5	74.7
>
> GaussianFusion-T 	77.6 75.0

---

> ### Author Response · Authors · 2025-11-20
> **Continue the above reply**
>
> Visualization
> APPENDIX Fig. 6 presents a visual comparison with the multi-modal fusion method OccFusion. The results show consistent performance between the multi-modal GaussianFusion and the camera-only GaussianFusion-C, demonstrating that GaussianFusion better distinguishes between different object categories (highlighted in red) and achieves a lower prediction error rate (indicated in yellow). This validates the effectiveness of the Gaussian-based multi-modal fusion framework.
>
> APPENDIX Fig. 7 illustrates a failure case, where both our GaussianFusion-C and the prior GaussianFormer miss parts of the drivable road surface, while the BEV-based SurroundOcc produces correct occupancy predictions (as marked in red). A possible explanation for this behavior lies in the inherent reliance of Gaussian representations on learned instance-aware density distributions. Unlike BEV-based methods, which explicitly model occupancy on a dense, regular grid—naturally capturing large-scale structures such as roads—Gaussian representations typically focus on compact, object-centric primitives. As a result, they may under-reconstruct spatially extended but texture-poor regions like road surfaces.

---

### Official Review · Reviewer_eFff · 2025-10-31

**Soundness:** 3
**Presentation:** 3
**Contribution:** 2
**Rating:** 6
**Confidence:** 3

**Summary:**

The authors presented GaussianFusion that replaces discrete BEV grids with a continuous 3D Gaussian space to fuse camera and LiDAR, aiming to preserve fine details and align modalities before any quantization. It introduces forward-projection camera Gaussian initialization, a shared Gaussian encoder with Gaussian-guided deformable attention and iterative property updates, then pools Gaussians to voxels for task heads; on nuScenes it reports +2.6 NDS over BEVFusion for detection and higher mIoU for occupancy with lower latency/memory

**Strengths:**

Continuous fusion before BEV: avoids early discretization loss and enables richer cross-modal interaction in Gaussian space.

Forward-projection init for cameras; shared encoder with Gaussian-guided deformable attention and incremental updates to µ/s/r.

Solid results + efficiency: NDS 74.0 / mAP 71.7 on val, latency 132 ms and memory 4271 MB vs. BEVFusion’s 156 ms/5140 MB; occupancy mIoU 28.65 SOTA among single-frame models.

**Weaknesses:**

Scope limited to nuScenes; no cross-dataset/domain validation shown, so generality beyond this setting is inferred rather than demonstrated.

Although fusion happens in continuous Gaussian space, the method must voxelize Gaussians for task heads, which can bring back quantization artifacts the paper set out to avoid.

At the same 200×200 BEV size, GaussianFusion shows higher memory than BEVFusion (5418 MB vs. 5140 MB), even though latency is lower.

Temporal modeling out of scope. Authors acknowledge that temporal fusion boosts SOTA in both detection and occupancy, but GaussianFusion leaves this to future work, so comparisons to temporal SOTA aren’t shown.

**Questions:**

Some questions are mentioned in the weakness section.

---

> ### Author Response · Authors · 2025-11-20
>
> 1. Scope limited to nuScenes
>
> We appreciate the insightful comment. To better assess generalization, we further conduct experiments on the Waymo dataset. The results show that Gaussian representation outperforms BEV-based methods, achieving higher accuracy and demonstrating greater potential.
>
> Model	mAP-L1	mAPH-L1	mAP-L2	mAPH-L2
>
> BEVFusion 82.72	81.35	77.65	76.33
>
> GaussianFusion（ours）	86.23	84.65	81.47	80.75
>
> 2. Although fusion happens in continuous Gaussian space…
> We adopted the same detection/occupancy head as BEVFusion, UniTR, and GaussianFormer to ensure a fair comparison—enabling us to isolate the benefits of Gaussian representation over BEV (Sec. 3.5, Perception Task Setup). This required voxelizing Gaussians for compatibility.
>
> To further validate our claim, we conduct an additional experiment using a query-based detection head (inspired by CMT), which operates directly on the Gaussian representation and avoids voxelization. Specifically, image and LiDAR Gaussians serve as KV inputs, with 3D position encoding applied consistently (as image Gaussians are also in 3D space). Queries are processed through a Transformer decoder with FFN to predict 3D boxes.
>
> Results show that this advanced query-based head outperforms BEVFusion’s heatmap-based approach—further supporting our argument that quantization artifacts in BEV representations can degrade performance, while continuous Gaussian modeling offers superior fidelity.
>
> detection head	NDS	mAP
>
> GaussianFusion（heatmap-based）	74.0	71.7
>
> GaussianFusion（query-based）	74.5(+0.5)	72.9(+1.2)
>
>
> 3. At the same 200×200 BEV size…
>
> The additional memory overhead of our method is minor—only ~200+ MB. More importantly, when increasing the BEV resolution from 200×200 to 400×400, GaussianFusion’s memory usage rises by just 13% (to 733 MB), whereas BEVFusion’s increases nearly 30× (to ~150,000 MB), leading to prohibitive computational cost—clearly unacceptable in practice.
>
> In contrast, GaussianFusion remains highly scalable. For instance, when extending the perception range from 50m to 100m in real-world autonomous driving scenarios, BEVFusion becomes infeasible due to memory explosion, while our method incurs only a 13% increase in memory consumption. This demonstrates GaussianFusion’s superior practicality and engineering viability for large-scale deployment.
>
> 4.Temporal modeling out of scope.
>
> We agree that temporal modeling is important. In the main paper, we focus on a frame-wise comparison with BEVFusion, UniTR, and GaussianFormer—without temporal fusion—to fairly isolate the benefits of Gaussian representation over BEV.
> To further validate the scalability of our approach, we introduce GaussianFusion-T, an extension with temporal modeling. Inspired by dense methods like BEVDet4D and BEVFusion4D—which warp historical BEV/3D features and apply deformable attention or 3D convolutions—we adapt our framework by warping historical Gaussian representations to the current timestamp and fusing them via our formulation (Eqs. 5–6).
>
> In this temporal setting, the variables in Eq. (5) include both current and past multi-modal Gaussians and queries.
> As shown in below, GaussianFusion-T outperforms BEVFusion4D by a large margin, and achieves competitive NDS compared to the state-of-the-art temporal sparse method SparseLIDAR-T—despite operating on a continuous, learned Gaussian representation.
>
> Method	Modality	NDS	mAP
>
> BEVFusion4D	C+L	73.5	72.0
>
> FusionFormer	C+L	74.1	71.4
>
> SparseLIF-T	C+L	77.5	74.7
>
> GaussianFusion-T	C+L	77.6	75.0
>
> This demonstrates that our framework not only enables fair and effective single-frame comparison, but also naturally extends to temporal modeling with strong performance—further highlighting the advantages of Gaussian-based scene representation.

---

### Official Review · Reviewer_RnZo · 2025-11-01

**Soundness:** 3
**Presentation:** 3
**Contribution:** 3
**Rating:** 6
**Confidence:** 4

**Summary:**

The paper introduces GaussianFusion, a new framework for multimodal 3D perception that replaces the traditional bird’s-eye view (BEV) grid representation with a 3D Gaussian representation. This approach unifies multimodal features (e.g., from different sensors) in a shared, continuous 3D Gaussian space, effectively preserving fine details and edges. The framework includes a forward-projection Gaussian initialization module and a cross-modal Gaussian encoder that iteratively refines Gaussian properties using attention mechanisms. GaussianFusion is task-agnostic and demonstrates strong effectiveness.

**Strengths:**

1. Clear and timely motivation. I initially thought BEV was the best in terms of representational capability and fusion suitability, but this work convincingly shows that 3D Gaussian representation is indeed a promising alternative.
2. The paper is well-structured and flows logically.
3. The proposed solution is intuitively convincing and demonstrates effectiveness in two visual representation use cases.

**Weaknesses:**

1. Missing ablation study on the impact of voxel size on task accuracy.
2. For readability, the citation format should be revised: use ~\citep or ~\citet instead of ~\cite.
3. Lacks discussion of failure cases where the Gaussian representation performs worse than the BEV representation.

**Questions:**

1. In Table 2, why is the modality listed as “C” when LiDAR (with VoxelNet backbone) is used?
2. Can this method be applied to anchor-aware object detection models (e.g., FUTR3D [1], TransFusion [2])?
3. Regarding Table 3, as far as I know, BEVFusion addresses long-latency issues for improved efficiency. Given that BEV-based methods quantize features early, I would expect them to have lower latency. However, GaussianFusion shows smaller latency than BEVFusion — where does this latency saving come from?
4. Could this approach generalize to other modality combinations (e.g., Camera + Radar, LiDAR + Radar, Camera + LiDAR + Radar)?
5. How effective is this approach on other datasets (e.g., Waymo) that have different sensor characteristics, such as varying LiDAR beam densities?

**Reference**

[1] FUTR3D: A Unified Sensor Fusion Framework for 3D Detection

[2] TransFusion: Robust LiDAR-Camera Fusion for 3D Object Detection with Transformers

---

> ### Author Response · Authors · 2025-11-20
>
> Weaknesses:
> 1. Missing ablation study on the impact of voxel size on task accuracy.
>
> We thank the reviewer for the comment. The detection performance under different voxel sizes is already provided in Column 3 of Table 1 in the main text. As shown, smaller grid sizes yield better performance (higher NDS) for both BEV-based and our Gaussian-based methods, indicating improved resolution benefits both representations.
> We further clarify this trend and now include mAP results across different grid sizes to provide a more comprehensive evaluation. These results consistently show that finer discretization improves detection accuracy, while our method maintains superior performance across settings.
>
> 2. For readability…
> We thank the reviewer for the suggestion. While the paper follows ICLR’s official formatting, we will release an ArXiv version with improved layout for better readability.
>
> 3. Lacks discussion of failure cases…
>
> APPENDIX Fig. 7 illustrates a failure case, where both our GaussianFusion-C and the prior GaussianFormer miss parts of the drivable road surface, while the BEV-based SurroundOcc produces correct occupancy predictions (as marked in red). A possible explanation for this behavior lies in the inherent reliance of Gaussian representations on learned instance-aware density distributions. Unlike BEV-based methods, which explicitly model occupancy on a dense, regular grid—naturally capturing large-scale structures such as roads—Gaussian representations typically focus on compact, object-centric primitives. As a result, they may under-reconstruct spatially extended but texture-poor regions like road surfaces.
>
> Questions:
>
> 1. In Table 2, why is the modality listed as “C” when LiDAR (with VoxelNet backbone) is used?
> This is a typo and should be empty. We will correct it to “-” in the final version.
>
> 2 .Can this method be applied to anchor-aware object detection models (e.g., FUTR3D [1], TransFusion [2])?
> We appreciate the insightful comment. Our framework can readily extend to anchor-based detectors such as FUTR3D, TransFusion, and CMT [1], with minimal modifications.
> As an example, we adapt CMT’s query-based head: image and LiDAR Gaussians serve as KV inputs, with 3D position encoding applied consistently (since image Gaussians are also in 3D space). Queries follow a standard Transformer decoder with FFN to predict 3D boxes.
> Results show that advanced query-based heads outperform BEVFusion’s heatmap-based head. Notably, our prior comparison with BEVFusion used its same detection head (Sec. 3.5, lines 298–303) for fairness.
> 3.Where does this latency saving come from?
> The speedup stems from our fully Gaussian-based scene representation. As in Table 5, we initialize 43,296 Gaussians, which are dynamically refined and reduced via Gaussian-to-Voxel Pooling. On average, only ~20k Gaussians remain at convergence—effectively halving the computational load.
> In contrast, BEV-based methods (e.g., 200×200 grid) require convolution over 40k fixed locations, leading to higher inference latency. Our adaptive, sparse Gaussian representation thus enables more efficient feature processing.
>
> 4. Could this approach generalize to other modality combinations (e.g., Camera + Radar…
>
> Yes, our framework can seamlessly support multiple sensors. Adding radar is straightforward—by designing a simple Gaussian initialization scheme, which can directly follow the LiDAR initialization due to their similar 3D geometry.
> We include a set of results (using an RCTrans-style detection head and extracting features from multi-modal Gaussians), showing that the Gaussian-based model achieves better performance.
>
> Method	NDS	mAP
>
> RC_BEVDet[2]	56.2	49.6
>
> RC_Trans[3]	59.4	52.0
>
> RC_GaussianFusion	62.3	54.6
>
> 5. How effective is this approach on other datasets.
>
> We appreciate the insightful comment. We further conduct experiments on the Waymo dataset, and the results demonstrate that Gaussian representation holds greater potential compared to BEV-based approaches.
>
> Model	mAP-L1	mAPH-L1	mAP-L2	mAPH-L2
>
> BEVFusion   82.72	81.35	77.65	76.33
>
> GaussianFusion（ours）	86.23	84.65	81.47	80.75
>
> Reference:
> [1]Cross Modal Transformer: Towards Fast and Robust 3D Object Detection
> [2] Bevdet: High-performance multi-camera 3d object detection in bird-eye-view.
> [3]RCTrans: Radar-Camera Transformer via Radar Densifier and Sequential Decoder for 3D Object Detection

---

> > ### Comment · Reviewer_RnZo · 2025-11-27
> >
> > Thanks for the detailed answer! My concerns are well addressed, and I’m happy to raise my score.

---

### Author Response · Authors · 2025-11-20

We sincerely thank the reviewers for their constructive feedback, which has been highly valuable in improving the quality of our paper. We have carefully addressed all comments to the best of our ability, clarified points of confusion, and strengthened the manuscript accordingly. All revisions in the main paper are highlighted in blue for easy reference.

Below, we provide a summary of the new content added to both the main text and the supplementary material:

1. Analysis and visualization of failure cases have been added to discuss limitations and provide insights (see Appendix Figure 7).
2. Camera-radar fusion experiments are included to validate the generalization of our framework to heterogeneous sensor modalities (see Appendix Table 10).
3. Experimental results on the Waymo Open Dataset are presented to demonstrate scalability and cross-dataset generalization (see Appendix Section F).
4. Temporal modeling with GaussianFusion-T and a discussion on future research directions are incorporated into the main text (see Table 4).
5. Experiments with a query-based detection head are conducted to evaluate model generality, compare with sparse detectors, and avoid re-quantization in Gaussian-to-voxel pooling (see Table 5).
6. Comprehensive analysis of computational cost in Gaussian-to-voxel conversion is provided to clarify efficiency and scalability (see Appendix Section B).
7. Differentiation of deformable attention with Gaussian features from prior works, along with ablation studies, are presented (see Figure 3 and Table 9 in the main text).
8. LiDAR-based Gaussian initialization and its formulation are detailed for reproducibility (see Appendix Section A).
9. Qualitative comparisons with the baseline OccFusion are added to highlight advantages in occupancy and structure recovery (see Appendix Figure 6).
10. An additional failure case with detailed explanation is included to further illustrate current limitations (see Appendix Figure 7).
11. Sensor dropout experiments are conducted to evaluate robustness under partial sensor input (see Appendix Tables 12 and 13).
12. Results for both single-task training and multi-task joint training are reported to clarify the training protocol and enable fair comparison (see 11. Appendix Table 14).

We believe these additions have significantly strengthened the paper, improved clarity, and addressed the reviewers’ concerns comprehensively.

---

> ### Author Response · Authors · 2025-11-25
> **I have uploaded my responses to all the reviewer comments. I would be grateful if the reviewers could review my replies at their earliest convenience.**
>
> I have uploaded my responses to all the reviewer comments. I would be grateful if the reviewers could review my replies at their earliest convenience.

---

### Meta-Review · Area_Chair_L7pu · 2025-12-19

**Summary:**

# Decision

The paper introduces 3D Gaussian-based framework for multimodal fusion, replacing discrete Bird’s-Eye View (BEV) representations.

The reviews are mixed. Reviewers praise the novel use of 3D Gaussian representations, solid technical contributions, and strong performance with notable speedups. However, they consistently criticize the limited and incomplete evaluation (single dataset), lack of ablations and qualitative analysis, and insufficient discussion of generalization and failure cases. Some also question the degree of novelty and methodological clarity, citing residual discretization, missing details, and unaddressed computational or memory overheads.

The authors, however, took the time to answer most of these main concerns, e.g., presenting results on a second dataset, introducing a discretization-free framework, answering computational questions, etc.

We believe that this updated submission could benefit the community, hence our recommendation for approval.

------------
# Consolidated Reviews

## Strengths

### Novel application of 3D Gaussian representation to multi-modal fusion.
- Promising alternative to classic BEV, leveraging Gaussian representation in a novel manner [`RnZo`, `eFff`, `x14q`, `mxcM`]
- Meaningful technical contributions (e.g., forward-projection for cameras) [`eFff`, `x14q`, `mxcM`]

### Significant results
- Convincing results w.r.t. object detection and occupancy prediction [`RnZo`, `eFff`, `x14q`]
- Significant 450% speedup [`x14q`]

### Misc.
- Clear, well-written [`RnZo`, `x14q`]

## Weaknesses

### Partial evaluation
- Evaluation on a single, dense dataset $\rightarrow$ lack of generalizability proof [`RnZo`, `eFff`, `x14q`]
- Incomplete comparison to sparse detectors [`x14q`, `mxcM`]
- Lack of discussion w.r.t. failure cases compared to traditional BEV [`RnZo`]
- Lack of ablation study w.r.t the impact of voxel size [`RnZo`]
- Limited qualitative results [`x14q`]

### Limitations in terms of novelty and methodology
- Lack of novelty for some components (e.g., encoder, deformable attention) [`x14q`]
- Discretization still happens, even though later  [`eFff`, `mxcM`]
- Glossed-over computational overhead of Gaussian-to-voxel conversion [`x14q`]
- Memory usage higher than baseline in some settings  [`eFff`]
- Technical details missing for some components [`x14q`]

### Misc.
- Inadequate citation format (`\cite`) [`RnZo`]

**Reviewer Concerns:**

See above for summary of main concerns shared by reviewers.

Overall, the authors have reasonably clarified the scope of their work and contributions. They also provide additional results on a second dataset, as well as experiments illustrating how their approach can be extended to a discretization-free, query-based framework, thus addressing several of the main concerns raised.

Nevertheless, the paper would still benefit from further clarification, particularly through more detailed discussion and visualization of the results, as well as from more precise writing.

**Reviewer Scores:**

### Reviewer `RnZo`
- **Original score:** 6
- **Score change:**  raised, likely to 8, c.f. reviewer's own reply.

### Reviewer `eFff`
- **Original score:** 6
- **Score change:** likely left unchanged, but small chance of being raised. The authors covered their concerns (evaluation on 2nd dataset, definition of discretization-free query-based solution) but at the costs of major changes in the paper's narrative.

### Reviewer `x14q`
- **Original score:** 4
- **Score change:** might have raised to 6 (or remained unresponsive). The authors decently covered the suggestions made the reviewer.


### Reviewer `mxcM`
- **Original score:** 2
- **Score change:** might have raised to 4 (or remained unresponsive). The authors did clarify some misunderstandings from the reviewer and provided a few additional results per the reviewer's request.

---

### Decision · Program_Chairs · 2026-01-26

Accept (Poster)